# Altered Tau Kinase Activity in rTg4510 Mice after a Single Interfaced CHIMERA Traumatic Brain Injury

**DOI:** 10.3390/ijms24119439

**Published:** 2023-05-29

**Authors:** Wai Hang Cheng, Honor Cheung, Amy Kang, Jianjia Fan, Jennifer Cooper, Mehwish Anwer, Carlos Barron, Anna Wilkinson, Grace Hu, Jefferey Yue, Peter A. Cripton, David J. Vocadlo, Cheryl L. Wellington

**Affiliations:** 1Department of Pathology and Laboratory Medicine, University of British Columbia, Vancouver, BC V6T 1Z4, Canada; chengwh@mail.ubc.ca (W.H.C.);; 2Djavad Mowafaghian Centre for Brain Health, University of British Columbia, Vancouver, BC V6T 1Z4, Canada; 3Department of Chemistry, Simon Fraser University, Vancouver, BC V5A 1S6, Canada; 4School of Biomedical Engineering, University of British Columbia, Vancouver, BC V6T 1Z4, Canada; 5International Collaboration on Repair Discoveries, Vancouver, BC V5Z 1M9, Canada

**Keywords:** traumatic brain injury, GSK-3β, tauopathy, CHIMERA

## Abstract

Traumatic brain injury (TBI) is an established risk factor for neurodegenerative diseases. In this study, we used the Closed Head Injury Model of Engineered Rotational Acceleration (CHIMERA) to investigate the effects of a single high-energy TBI in rTg4510 mice, a mouse model of tauopathy. Fifteen male rTg4510 mice (4 mo) were impacted at 4.0 J using interfaced CHIMERA and were compared to sham controls. Immediately after injury, the TBI mice showed significant mortality (7/15; 47%) and a prolonged duration of loss of the righting reflex. At 2 mo post-injury, surviving mice displayed significant microgliosis (Iba1) and axonal injury (Neurosilver). Western blotting indicated a reduced p-GSK-3β (S9):GSK-3β ratio in TBI mice, suggesting chronic activation of tau kinase. Although longitudinal analysis of plasma total tau suggested that TBI accelerates the appearance of tau in the circulation, there were no significant differences in brain total or p-tau levels, nor did we observe evidence of enhanced neurodegeneration in TBI mice compared to sham mice. In summary, we showed that a single high-energy head impact induces chronic white matter injury and altered GSK-3β activity without an apparent change in post-injury tauopathy in rTg4510 mice.

## 1. Introduction

Traumatic brain injury (TBI) is a leading worldwide cause of death and disability. TBI is clinically classified as mild, moderate, or severe, based on the Glasgow Coma Scale and additional clinical indicators, including loss of consciousness and post-traumatic amnesia. Repetitive exposure to mild TBI (mTBI) is associated with increased risk of developing a neurodegenerative condition called chronic traumatic encephalopathy (CTE) [1,2,3]. The pathognomonic features of CTE include aggregates of perivascular phosphorylated tau (p-tau), located at the depths of the cortical sulci [4]. In addition to repetitive mTBI, several reports and case studies have shown that a single exposure to moderate or severe TBI (msTBI) may also induce p-tau neuropathology.

A previous report comparing fatal TBI cases to controls (*n* = 5 per age group of <20 years, 20–50 years, >50 years) identified subtle tau immunoreactivity in glial cells during the acute period (<24 h to 1 month) after msTBI [5]. A recent study of 39 post-mortem brains of chronic survivors (1–47 years post-injury) of a single msTBI found that, in the young cohort, tau-positive neurofibrillary tangles (NFTs) were more frequently observed in TBI cases than in age-matched controls [6]. Other case studies [7,8] examined post-mortem brains from patients who survived decades (24 to 42 years post-injury) after a single msTBI (mostly due to motor vehicle accidents or gunshots) and observed p-tau immunoreactivity and NFTs at perivascular sites, the superficial and deep cortical layers (sulcal depths), the hypothalamus, and the brainstem. Notably, there was also evidence of multiple proteinopathies, including TDP43. Another recent study [9] used positron emission tomography (PET) with the tau tracer flortaucipir to study tau pathology in 21 patients with a remote history of a single msTBI (range 18–51 years post-injury, median 32 years), compared to age-matched controls, and observed increased flortaucipir-binding at the right lateral occipital cortex in patients with a history of msTBI. In addition, flortaucipir binding in TBI patients was associated with increased total tau and p-tau concentrations in cerebrospinal fluid (CSF). Other case studies have examined the post-mortem brains of schizophrenia patients who underwent prefrontal leucotomy [10,11]. These cases represent a human model of long-term (more than 40 years post-injury) single traumatic axonal injury. These studies observed perivascular p-tau, NFTs, and astrocytic tangles only in leucotomized brains. Together, these studies suggest that a single msTBI may be sufficient to induce long-term tau pathology. In addition, p-tau pathologies may be triggered by axonal injury in general and are not necessarily limited to mechanical impact.

Many experimental studies have studied the effects of TBI in various transgenic mouse models that harbor the human *MAPT* gene. Most of these studies have focused on relatively “mild” TBI and focused primarily on acute and sub-acute post-injury outcomes. Yoshiyama et al. [12] induced repetitive closed head impacts (16×, 4 times per day, 1 day per week) to 12-mo Tg T44 mice, overexpressing the shortest human tau isoform T44. At 9-mo post-TBI, only 1 of 18 injured mice showed extensive NFT development at the hippocampus and entorhinal cortex and increased p-tau (PHF1, PHF6, 12E8). Other groups have investigated TBI effects in mice that lack endogenous mouse tau and express all six isoforms of human tau (hTau mice), with inconsistent results. Specifically, Mouzon et al. [13] induced repetitive closed head impact (5× over 9 days) in 3-mo or 12- to 13-mo hTau mice and observed increased p-tau (RZ3) in the hippocampus at 24 h post-injury in male, but not female, mice. Gangoli et al. [14] used CHIMERA without interface and performed repetitive mTBI (20 × 0.24 J or 20 × 0.13 J) in 4-mo hTau mice and observed no p-tau in TBI mice at 1 yr post-injury. Ojo et al. [15] induced repetitive closed head impacts in 12-wk-old hTau mice (24 impacts over 3 months or 32 impacts over 4 months), with an inter-injury time of 72 to 96 h, and they observed increased total tau, tau oligomers (TOC-1), and p-tau (pThr231) at 3 mo after the final injury in hTau mice. Other researchers have studied the effects of TBI in mouse models that express mutant forms of tau found in neurodegenerative disease. Tran et al. [16] performed control cortical impacts (CCIs) at 3 severities (mild, mild-moderate, moderate) in 5- to 7-mo 3xTg-AD mice and observed increased tau punctae and increased p-tau (pS199, pT205, pT231, pS396, pS422, AT8, AT100, AT180) in the hippocampus, fimbria, and amygdala of the “moderate” injury group from 24 h to 7 d post-TBI. Xu et al. [17] performed weight drop TBI (1×, 4× over 7 days or 12× over 7 days) in 5-wk-old mice overexpressing P301S tau (1N4R) and observed an injury dose-dependent increase in p-tau (pS422) in the retinas of TBI mice. Using the same P301S tau model, Edwards et al. [18] induced a single msTBI using CCI in 3-mo P301S mice at the right parietal cortex. They observed increased p-tau (AT8) in both the ipsilateral and the contralateral cortices and the hippocampus up to 6-mo post-injury. Interestingly, they also observed tau spreading to the brainstem and possibly the hypothalamus months after injury. Cheng et al. [19] induced rmTBI using closed head impact (42 impacts over 7 days) in 3- to 4-mo Tau58.4 mice overexpressing P301S human tau and did not observe altered tauopathy up to 3 mo post-injury. Bachstetter et al. [20] induced 1 or 2 closed head impacts 24 h apart in 2-mo rTg4510 mice that overexpressed the P301L human tau (0N4R) and observed increased p-tau (pS396/pS404) by 15 days after the first injury. The discrepancies across these studies likely stem from the many variables in study designs, including those focused on the TBI itself (i.e., TBI model, injury severity, injury frequency), as well as in modeling tauopathy (wildtype human tau vs. mutant tau, overexpression vs. endogenous levels, and ratios of different tau isoforms). In general, TBI appears to have a greater probability of inducing tauopathy in mouse models overexpressing mutant forms of tau.

We developed an impact-acceleration model of TBI known as CHIMERA [21], which allows for unrestrained head motion during impact. We recently published a modification that allows us to deliver a high impact energy injury to mice using an interface to distribute impact energy across the skull, thereby avoiding skull fracture, to mimic more severe TBIs [22,23]. In C57/Bl6 mice, we showed that interfaced TBI at 2.5 J resulted in a mortality of <20%, neurological and memory deficits, elevated plasma total tau and neurofilament light levels, increased brain cytokine levels, and blood–brain barrier disruption [22]. The present study was designed to evaluate the effect of a single high impact-energy TBI on chronic tauopathy in rTg4510 mice. Here we used interfaced CHIMERA to induce a single TBI in 4-mo rTg4510 mice using 4.0 J of impact energy, which produced an overall mortality of 46.7%. Animals were aged for 2 mo before brain tissues were harvested for histological and immunoblotting analyses. Blood samples were collected for plasma biomarker measurements. We observed that a single high-energy TBI was sufficient to induce long-term white matter injury and prolonged activation of GSK-3β. However, tauopathy and neurodegeneration were not significantly different between the sham and TBI animals. These findings suggest that a single high-energy head impact may be sufficient to induce long-term white matter injury and increased activity of tau kinases but without a marked effect on tauopathy or neurodegeneration.

## 2. Results

### 2.1. CHIMERA TBI with Interface at 4.0 J Induced High Mortality

We first performed a pilot impact energy titration experiment to establish CHIMERA conditions to mimic a moderate-severe TBI in 4-mo rTg4510 mice, overexpressing the P301L human 4R0N tau. Interfaced CHIMERA TBI impact energies ranging from 2.3 J to 5.8 J were tested. Survival was 100% in mice impacted at 2.3–2.9 J, 100% at 3.5–3.8 J, 53.8% at 4.0 J, 50.0% at 4.2 J, and 0% at 4.5–4.8 J (Appendix A). To produce experimental TBI with an *a priori* defined overall mortality rate of ~50%, we selected an impact energy range of 4.0 J for the full study.

### 2.2. TBI Induced Much Prolonged Loss of Righting Reflex and Chronic Axonal Injury and Microgliosis

For the full study, seven rTg4510 mice were randomized to the sham group, which received all procedures except for impact, and 15 rTg4510 mice were randomized to the TBI group and impacted at 4.0 J (*n* = 15). Overall, eight TBI mice survived. Non-surviving mice either died immediately after TBI or reached the humane end point soon thereafter, resulting in an overall mortality rate of 46.7% (7/15) (Figure 1A, Appendix A). The duration of loss of the righting reflex (LRR, analogous to loss of consciousness in humans) in all surviving mice is shown in Figure 1B. In addition, one TBI mouse that survived the procedure and regained consciousness but reached the humane end point at 6 h (LRR value = 1680) is also included in the figure. On average, sham-operated mice had an LRR duration of 140 s, whereas TBI mice had a significantly longer LRR duration (2225 s, *p* = 0.0002).

After sham or TBI procedures, the eight surviving rTg4510 mice were aged for 2 months before brain samples were harvested at 6 months of age. Compared to sham controls, TBI mice had chronic microgliosis at the optic tract, as revealed by Iba1 immunohistochemistry (*p* = 0.0082), as well as axonal injury in the optic tract, as revealed by silver staining (*p* = 0.0012) (Figure 1C). These results are consistent with our previous studies demonstrating chronic white matter injury, particularly in the optic tract, after interfaced CHIMERA impacts in C57Bl/6 mice [22,24].

### 2.3. TBI Induced Chronic Activation of GSK-3β

Since rTg4510 mice express high levels of 4R P310L human tau, we tested whether TBI induced changes in the levels of tau kinases. We first performed Western blotting of RIPA lysates to probe for different forms of GSK-3β, a major Ser/Thr kinase responsible for tau phosphorylation. Although TBI did not affect the level of total GSK-3β (*p* = 0.6499), TBI significantly reduced the ratio of phosphorylated GSK-3β (p-GSK-3β S9) to total GSK-3β (*p* = 0.0132) (Figure 2A). Since p-GSK-3β S9 is an inactive form of GSK-3β, this finding suggests that GSK-3β activity was chronically enhanced in these animals. We then investigated AKT (also known as protein kinase B), which is a Ser/Thr kinase and a major regulator of GSK-3β signaling. When activated, AKT phosphorylates GSK-3β at S9, leading to its inhibition. Western blotting of RIPA lysates revealed no significant change in total AKT levels (*p* = 0.9979). However, there was a strong trend toward a reduced ratio of p-AKT (S473) to total AKT (*p* = 0.0582) (Figure 2B). We conclude that high impact-energy TBI in rTg4510 mice chronically activated GSK-3β, potentially due to disrupted AKT-GSK-3β signaling.

### 2.4. TBI Accelerated Increase in Plasma Total Tau in rTg4510 Mice

We collected longitudinal plasma samples from sham and TBI mice starting from baseline (2 weeks before TBI or sham operation), at 6 h post-TBI, and every 2 weeks thereafter. Plasma total tau (t-tau) was assayed using a human t-tau SIMOA assay. Despite loss of some samples due to machine error, we observed the expected time-dependent increase in t-tau from 2 to 8 weeks post-operation (Figure 3A). We then proceeded to analyze sham vs. TBI differences using four-parameter logistic regression (Figure 3B). The best fitted models of the sham and TBI groups did not differ greatly in terms of the upper plateau (10^2.567^ = 369 pg/mL vs. 10^2.733^ = 541 pg/mL). Intriguingly, we noted that the sham group exhibited a point of inflection at 18 days post-sham procedure (10^1.506^ = 32 days post-baseline), whereas the TBI group had a point of inflection within 1 day post-TBI (10^1.170^ = 14.8 days post-baseline). This finding suggests that interfaced TBI at 4.0 J may accelerate the increase in plasma t-tau in rTg4510 mice by approximately 2 weeks, although it may not affect the final level of plasma tau at 2-mo post-TBI.

### 2.5. TBI Did Not Change Brain Tau Burden in rTg4510 Mice at 2-mo Post-Injury

Since we observed increased brain GSK-3β activity and a possible acceleration of the elevation of t-tau in plasma, we next sought to evaluate whether TBI induced tau burden in rTg4510 mice, using immunohistochemistry, histochemical staining, and Western blotting (Figure 4A–F). Overall, TBI did not induce significant changes in the level of total tau (DA9, Figure 4A) or neurofibrillary tangles (NFT, by Gallyas Silver stain, Figure 4B). We next performed IHC using antibodies that recognize different phosphorylated tau epitopes: PHF1 (pS396+pS404), AT8 (pS202+pT205), CP13 (pS202), and MC1 (pathological conformation) (Figure 4C–F, Appendix A). When we compared sham vs. all TBI animals, we did not observe a significant increase in any p-tau epitope in any brain region examined (dorsal hippocampus, ventral hippocampus, entorhinal cortex, frontal cortex, amygdala, optic tract, hypothalamus). These findings suggest that, despite increased GSK-3β activity in the brain and accelerated elevation of total tau in plasma, the tau burden in rTg4510 was not increased at 2-mo post-injury.

### 2.6. TBI Did Not Change Levels of Autophagosomes and Lysosomes, Neurons, Astrocytes, Endothelial Cells, Synapses, or Size of Brain Regions

We then investigated whether the changes in protein degradation pathways may explain the lack of significant change in tau burden in the brain. We probed for autophagosome adaptor protein (p62), autophagosome initiation proteins (LC3B-I and LC3B-II), and lysosomal proteases (cathepsin B and cathepsin D) using IHC or Western blotting of RIPA lysates. No significant differences between sham and TBI mice were observed (Appendix A).

We also performed IHC analysis using NeuN, GFAP, CD31, and IgG antibodies to stain for neurons, astrocytes, endothelial cells, and extravasated IgG, respectively. We observed no significant change when comparing sham vs. TBI brains across any region analyzed (Figure 5A–C). Western blotting analysis also showed no differences in synaptic markers (synaptophysin and PSD95) between sham vs. TBI mice (Figure 5D).

We also analyzed the size of various brain regions and ventricles in sham vs. TBI animals (Figure 5E). The sizes of the dorsal hippocampus and amygdala were measured on the coronal plane (approximately −2 mm posterior to the bregma), and the area of the lateral ventricle was measured on the coronal plane of the frontal cortex (approximately 1.2 mm anterior to the bregma). There was no significant difference in the size of these regions in sham vs. TBI animals, suggesting no effect of TBI on neurodegeneration.

## 3. Discussion

This study was designed to investigate the outcomes of a single high-energy head impact induced by interfaced CHIMERA in a mouse model of tauopathy. At an impact energy of 4.0 J, designed to mimic a moderate-severe TBI, the overall mortality rate in TBI mice was 46.7%. This mortality rate is vastly different from previous CHIMERA studies (0 mortality at 0.1-J to 0.7-J impacts without interface [21,24,25,26,27,28]; 9–20% morality at 2.5-J impacts with interface [22,23]). The mortality rate in the current study is well aligned with clinical observations of msTBI, in which mortality rate can be as high as 43–46% [29,30,31]. In addition, the high-energy head impacts in this study induced a much longer duration of LRR in surviving mice (median >30 min in this study, compared to ~5 min in non-interface 0.1- to 0.7-J CHIMERA TBI [21,24,25,26,27,28] and ~15 min in 2.5-J interfaced TBI [22,23]).

The most important finding in this study is that TBI chronically activates GSK-3β, which is one of the most well-studied tau kinases, and it phosphorylates tau at multiple sites [32,33,34]. Its activation has been found in NFTs of AD patients [34], and its inhibition is an active area of therapeutic research for neurodegenerative diseases [35]. In rodent models of TBI [36,37,38,39], increased GSK-3β phosphorylation at serine 9 (which inactivates GSK-3β activity [40]) has been observed during the acute post-injury phase and may be important to neuronal survival in the early period. However, the role of GSK-3β in the chronic post-injury phase has not been reported. Our study suggests that TBI chronically disrupts AKT-GSK-3β signaling, leading to increased GSK-3β activity. This finding highlights the potential benefits of inhibiting or competing against GSK-3β activity after TBI in chronic management.

At 2 mo post-injury, we observed no significant change in tau burden. Although we confirmed that tau expression remained unchanged (Appendix A), we noted that the TBI group showed increased variability across several measures. Specifically, the coefficient of variation (CoV) of tau and p-tau across multiple brain areas in sham mice is ~50%, but it was 90% in the TBI group. Similar observations were found in ventricle sizes. Therefore, it is possible that variations in post-injury tau clearance efficiency may have masked the effects of tau kinase activity. Using Iba1 as microglia marker, we observed no significant differences in the cortical microglial response. Future studies could use phagocytic markers, such as CD68 and Trem2 [41,42], to determine whether microglia showed potential phagocytic differences. In addition, we used p62, LC3B, cathepsin B, and cathepsin D to investigate autophagolysosomes, a major cellular degradation pathway of tau [43,44,45], and we did not observe any significant differences between sham and TBI animals (Appendix A). Notably, p62 and CTSD levels were significantly and positively correlated with cortical tau levels, with a subset of TBI samples with very high p62 levels (Appendix A). This observation is aligned with previous reports suggesting that TBI may impair autophagy flux [46]. Future studies could investigate whether variations in post-injury autophagy responses contribute to tau clearance.

To our knowledge, we are the first to report longitudinal plasma total tau changes in a mouse model of tauopathy, and we report that the temporal elevation of plasma tau can be accelerated by TBI. However, we did not observe a significant change in brain tau burden at 2 mo post injury, which could be due to the following limitations. First, for brain tauopathy, we had only one time point of observation. Second, our study had a small N and high mortality, which could have induced potential survivor bias in the TBI group. Third, it is unclear how plasma total tau reflects brain tau levels. Nevertheless, the observation that TBI appears to accelerate a shift to high plasma tau levels is potentially interesting, and future studies could incorporate additional blood biomarkers.

In conclusion, a single high impact-energy TBI delivered to rTg4510 mice using the CHIMERA platform resulted in extended loss of the righting reflex, chronic white matter injury, increased GSK-3β activity, and potential acceleration of the elevation of plasma total tau levels.

## 4. Materials and Methods

### 4.1. Animals

All experiments were approved by the University of British Columbia Animal Care Committee and were compliant with the Canadian Council of Animal Care (A15-0096). Male rTg4510 mice (Jackson Laboratory #024854) were purchased from the Jackson Laboratory (Bar Harbor, ME, USA). These mice express a tetracycline-regulatable tetO-MAPT*P301L transgene under the control of the murine prion protein (PrP) promoter, leading to overexpression of human four-repeat (4R0N) mutant P301L tau. These mice also harbor the CaMK2a-tTA transgene, which suppresses tau expression upon exposure to tetracycline or its analogs. Tetracycline was not used in this study. The mice were housed with environmental enrichment on a 12-h/12-h reversed light cycle and received the 2918 Teklad Global 18% Protein rodent diet (Inotiv, Madison, WI, USA) and autoclaved reverse osmosis water *ad libitum*.

### 4.2. Traumatic Brain Injury

At 4 mo (133.4 ± 0.5 days) of age, male rTg4510 mice received a single interfaced TBI using the CHIMERA device, as previously described [22]. Immediately prior to TBI, the mice received 0.5 mL of NaCl for fluid supplementation and 1 mg/kg of meloxicam for analgesia. Anesthesia was induced using 5% isoflurane at 2.5 L/min oxygen and thereafter maintained at 3–4%. Anesthetized mice were restrained by abdominal straps in the supine position on the CHIMERA device such that their heads were free to move and rested at an angle of approximately 145° relative to the body. A polylactic acid (PLA)-silicone interface fitted to the contour of the mouse skull was placed under the animal’s head to protect it from skull fracture and distribute impact energy evenly across the skull. Head acceleration and rotational motion were on the sagittal plane. For sham controls, mice received fluid supplementation, analgesia, anesthesia, and positioning in the CHIMERA device but no impact. During the TBI procedures, the duration of isoflurane exposure and loss of consciousness were recorded. Impact energies ranged from 2.3 J to 5.8 J in a pilot experiment designed to define the maximum tolerable impact energy, and an impact energy of 4.0 J was used for the remainder of this study. Chest compressions with oxygen supplementation were performed on all mice that experienced cardiac/respiratory arrest immediately after the TBI procedure. Most of the animals that did not survive the procedure died within seconds to minutes post-injury. A summary of mortality and causes of death is provided in Appendix A.

### 4.3. Blood Collection and Euthanasia

Longitudinal blood (EDTA–plasma) samples were collected from the saphenous vein 1 week before TBI (defined as baseline), at 6 h after TBI, and every 2 weeks thereafter, using capillary collection tubes (ThermoFisher, Waltham, MA, USA). At 2 months post-TBI (6 mo of age), animals were euthanized with 150 mg/kg ketamine (Zoetis, Parsippanny, NJ, USA) and 20 mg/kg xylazine (Bayer, Creve Coeur, MO, USA). Cardiac puncture was performed to collect terminal blood samples. The mice were then perfused with 50 mL of ice-cold heparinized PBS (5 USP unit/mL). The mouse brain was dissected longitudinally, and 1 hemibrain was frozen for protein homogenization, while the other half was fixed in 4% paraformaldehyde (PFA) for histology. All blood samples were centrifuged at 1000× *g* for 10 min and the supernatant was stored at −80 °C as EDTA–plasma.

### 4.4. Histology, Immunohistochemistry and Immunofluorescence

Hemibrains were fixed in 4% PFA for 2 days and cryoprotected with 30% PBS–sucrose for 3 days, after which 40 μm-thick coronal sections were cut using a cryotome (Leica, Wetzlar, Germany). Injured axons were stained using the NeuroSilver Staining Kit (FD NeuroTechnologies, Columbia, SC, USA) following the manufacturer’s instructions. Neurofibrillary tangles were stained using the Gallyas Silver stain protocol, adapted from [47,48].

Immunohistochemistry (IHC) for the microglial marker Iba1 was performed as described [21]. Briefly, sections were quenched with hydrogen peroxide for 10 min, blocked with 5% BSA, and incubated with Iba1 antibody (Wako 019-19741, Richmond, VA, USA) overnight at 4 °C. Sections were then incubated with biotin-conjugated anti-rabbit secondary antibodies (Vector, Vectba1000 1:1000, Newark, NJ, USA) for 2 h at RT and then with ABC reagent (Vector, PK-6100, 1:400, Newark, USA) for 1 h before color development with 3,3′ diaminobenzidine (DAB) (Sigma D5637, Burlington, VT, USA). For IHC of tau, total tau was detected with DA9 (1:50,000), and p-tau was detected with AT8 (pS202/T205; ThermoFisher MN1020, 1:500, Waltham, MA, USA), CP13 (pS202; 1:5000), PHF1 (pS396/S404; 1:5000), and MC1 (pathological conformation; 1:500). DA9, CP13, PHF1, and MC1 were generous gifts from Dr. Peter Davis. Briefly, sections were quenched with hydrogen peroxide for 30 min, blocked with 5% BSA, and incubated with primary antibody overnight and then anti-mouse IgG1 (Southern Biotech 1071-08, 1:1000, Birmingham, AL, USA) in 20% Superblock (ThermoFisher PI-37535, Waltham, MA, USA) for 2 h. The sections were then incubated with ABC and developed with DAB as above.

Immunofluorescence of the neuronal marker NeuN was performed by permeabilization of sections with 0.1% Triton X-100 for 30 min, blocking with 5% normal goat serum (NGS), incubation with NeuN antibody (Abcam 104225, 1:1000, Waltham, MA, USA) overnight at 4 °C, and incubation with anti-rabbit secondary antibody (ThermoFisher A11-012, 1:500, Waltham, MA, USA) at RT for 2 h. For the autophagosome cargo protein p62, sections were permeabilized, blocked, and probed with p62 antibody (NEB D5L7G, 1:800, Ipswich, MA, USA) and streptavidin-594 (Biolegend 405240, 1:500, San Diego, CA, USA), using the Mouse on Mouse kit (Vector BMK2202, Newark, USA) per the manufacturer’s protocol. The lysosomal proteins cathepsin B and cathepsin D were detected with the antibodies NEB DIC7Y (1:1000) and NEB E179 (1:250), respectively. Detection of mouse immunoglobulin G (IgG) was performed using IgG-cy3 antibody (Jackson Immuno Research 715-165-150, 1:50, Philadelphia, PA, USA). Co-staining of astrocytes and endothelial cells was performed with antigen retrieval (boiling in sodium citrate buffer in a pressure cooker for 5 min) and probing with GFAP-488 (eBioscience 53-9892-80, 1:400; San Diego, CA, USA) and CD31 (Abcam 28364, 1:200, Waltham, MA, USA).

### 4.5. Image Quantification

All coronal brain histology sections were imaged using an Axio Scan.Z1 slide scanner (Zeiss, Oberkochen, Germany) with a 20× objective. Regions of interest (ROIs) included the olfactory bulb, frontal cortex, retrosplenial cortex, hippocampus (dorsal and ventral), amygdala, corpus callosum, optic tract, hypothalamus, and ventricles. Quantification of signals from Iba1, NeuroSilver, Gallyas Silver, PHF1, AT8, CP13, MC1, p62, NeuN, and IgG was performed by thresholding and reporting %Area (=Signal Area/ROI Area × 100%) [21,24]. Quantification of GFAP and CD31 signals was performed by reporting the mean signal intensity over the ROI and % overlapping area as previously described [23]. Quantification of CTSB and CTSD signals was performed by reported integrated density over ROI.

### 4.6. Tissue Homogenization and Western Blot

Proteins from half-brain samples were serially extracted as described [49]. Briefly, tissues were homogenized in 30 μL/mg of RAB buffer (100 mM MES, 1 mM EDTA, 0.5 mM MgSO_4_, 750 mM NaCl, 20 mM NaF, 1 mM Na_3_VO_4_), supplemented with protease inhibitor (Roche, Basel, Switzerland) and phosphatase inhibitor (Roche, Basel, Switzerland). The homogenate was centrifuged at 50,000× *g* for 20 min at 4 °C, and the supernatant was collected as the RAB soluble fraction. The pellet was resuspended in 30 μL/mg RIPA buffer (150 mM NaCl, 50 mM Tris, 0.5% deoxycholic acid, 1% Triton X-100, 0.5% SDS, 25 mM EDTA pH 8.0), supplemented with protease inhibitor (Roche, Basel, Switzerland) and phosphatase inhibitor (Roche, Basel, Switzerland), and centrifuged at 50,000× *g* for 20 min at 4 °C, and the supernatant was collected as the RIPA soluble fraction.

Immunoblots were performed by resolving 10 μg of RAB or RIPA fraction through 10% denaturing SDS-PAGE and then transfer to polyvinylidene fluoride (PVDF) membranes (Millipore, Burlington, VT, USA). Blots were probed by primary and secondary antibodies, developed with Supersignal West Femto (Thermo, Waltham, MA, USA), and captured by a ChemiDoc MP imaging system (Biorad, Hercules, CA, USA). The primary antibodies used were: DA9 (1:1,000,000), RZ3 (1:50,000), GluN1 (NEB D65B7, 1:10,000, Ipswich, MA, USA), Synaptophysin (Abcam YE269, 1:50,000, Waltham, MA, USA), PSD95 (Abcam 18258, 1:100,000, Waltham, MA, USA), p-GSK-3β (S9) (NEB D85E12, 1:50,000, Ipswich, MA, USA), GSK-3β (NEB 27C10, 1:100,000, Ipswich, MA, USA), and GAPDH (Millipore MAB374, 1:100,000, Burlington, VT, USA). DA9, RZ3, and PHF1 were generous gifts from Dr. Peter Davis. Densitometry was quantified with ImageJ (NIH) software using GAPDH for normalization.

### 4.7. Plasma Total Tau Analysis

Plasma samples from 17 mice were analyzed with the Quanterix^®^ Simoa HD-1^®^ analyzer. Samples were analyzed with the Simoa Tau Advantage Kit (101552) using the manufacturer’s protocol. Plasma samples were diluted off board at a 50-fold dilution using the sample diluent provided. Longitudinal samples were collected from all mice over 6 time points (baseline, 6 h, 2 wk, 4 wk, 6 wk, 8 wk). However, some samples were lost due to machine error. All remaining samples (*n* = 64) were randomized and analyzed on a single plate using the provided 8-point calibrator curve and 2 controls. The curve had an average percentage error of 11% and an average recovery of 100%. Both controls were within acceptable ranges as specified by the manufacturer. *n* = 52 samples were run in duplicate, with an average CV of 11%. *n* = 11 samples from 8 mice were run singly due to volume constraints. *n* = 7 samples were greater than the upper limit of quantification but within the limits of detection. No samples were less than the lower limits of quantification.

### 4.8. Animal Genotyping

DNA was extracted from ear notches using QIAamp DNA mini kit (Qiagen 51306, Hilden, Germany) following the manufacturer’s instructions. To genotype the Camk2a-tTA transgene, a standard polymerase chain reaction (PCR) assay was performed following the Jax Genotyping Protocol #18439. Specifically, PCR Master Mix (ThermoFisher K0171, Waltham, MA, USA) and the following cycling conditions were used on a Bio-Rad T100 Thermal Cycler: initial denaturation at 95 °C for 3 min followed by 36 cycles of amplification at 95 °C (30 s), 56 °C (30 s), and 72 °C (1 min), with a final annealing step at 72 °C for 5 min. Samples were kept at 4 °C until run on 1.5% agarose gel (130 V). The amplicon size for the transgene is ~550 base-pairs (bp), whereas the internal positive control yields a band of 324 bp. To genotype the transgene MAPT, probe-specific real-time quantitative PCR (qPCR) was performed using LightCycler^®^ Multiplex DNA Master (Roche07339585001, Basel, Switzerland) on a LightCycler^®^ 96 system (Roche, Basel, Switzerland) following Jax Genotyping Protocol #22109. qPCR was also used to quantify gene doses of tTa and MAPT in genomic DNA using DNA Green reagents (Roche 06402712001, Basel, Switzerland). The primer sequences for tTa were forward: 5′-GGA CGA GCT CCA CTT AGA CG-3′; and reverse: 5′-CAA CAT GTC CAG ATC GAA ATC 3′. The same sequences of MAPT and the internal control (APOB) from Jax Protocol #22109 were used to assess gene dose.

### 4.9. Quantitative Reverse Transcriptase-PCR (qRT-PCR)

We used quantitative polymerase chain reaction (qPCR) to assess transgene dosage using 50 genomic DNA and transgenic expression levels using mRNA. RNA was extracted using a PureLink^TM^ RNA mini-kit (ThermoFisher 12183018A) followed by cDNA synthesis using TaqMan^TM^ Reverse Transcription Reagents (ThermoFisher N808-0234) according to the manufacturers’ protocols. Quantitative RT-PCR was performed with DNA Green reagents (Roche 06402712001) on a LightCycler^®^ 96 system (Roche, Basel, Switzerland). Each sample was assayed in duplicate and normalized to glyceraldehyde 3-phosphate dehydrogenase (GAPDH). The following primer sequences were used: mouse *Gapdh* forward 5′-AAG GTC ATC CCA GAG CTG AA-3′, reverse 5′-CTG CTT CAC CAC CTT CTT GA-3′; mouse *Actb* (Actin) forward 5′-ACG GCC AGG TCA TCA CTA TTG-3′, reverse 5′-CAA GAA GGA AGG CTG GAA AAG-3′; human MAPT (Tau) forward 5′-CCC AAT CAC TGC CTA TAC CC-3′, reverse 5′-CCA CGA GAA TGC GAA GGA-3′; and tTa forward 5′-GGA CGA GCT CCA CTT AGA CG-3′, reverse 5′-CAA CAT GTC CAG ATC GAA ATC-3′.

### 4.10. Statistics

In this study, most analyses (loss of righting reflex, histology, immunohistochemistry, and Western blot) were comparisons between 2 groups (sham and TBI). Data were analyzed by Student’s *t*-test (if normally distributed) or the Mann–Whitney U test (if not normally distributed). The statistical method used for each analysis is stated in the figure legend. Longitudinal measurements of plasma total tau were analyzed by log tau level vs. log time, using a 4-parameter logistic curve model. The effect of impact energy on mortality was modeled using a logistic regression model. Correlational analyses of tau vs. autophagolysosomal markers were performed using Pearson’s correlation.

## Figures and Tables

**Figure 1 ijms-24-09439-f001:**
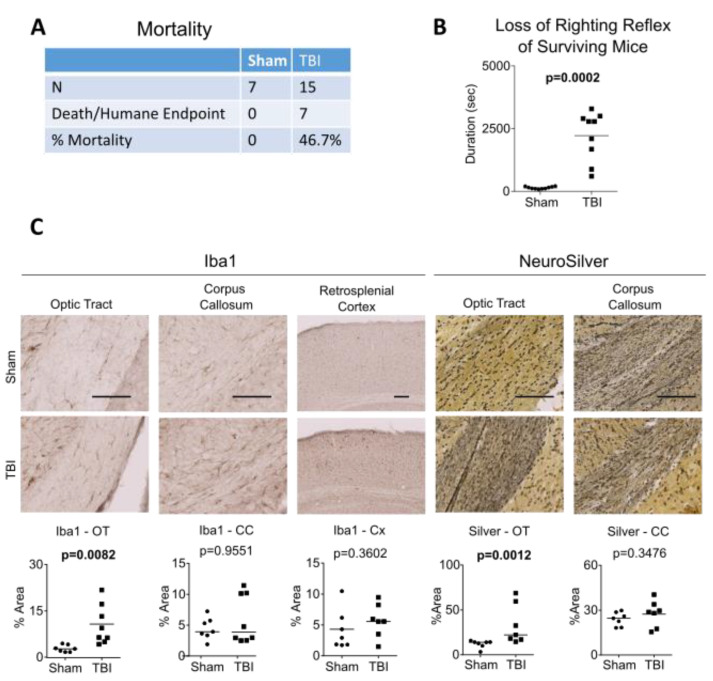
Mortality, loss of righting reflex, axonal injury and white matter microgliosis in rTg4510 after interfaced TBI. (**A**) Mortality rate of mice after a single TBI at 4.0 J with interface. (**B**) Duration of loss of righting reflex of sham and TBI animals. (**C**) White matter injury in sham and TBI animals as assessed by Iba1 immunohistochemistry in microglia and NeuroSilver staining for degenerative axons. Results are quantified in the graphs below the images. Scale bar = 100 μm. The Mann–Whitney U test was used for LRR, Iba1-CC, and Silver-OT, where horizontal lines indicate group median. The *t*-test was used for all other analyses, where horizontal lines indicate the group mean.

**Figure 2 ijms-24-09439-f002:**
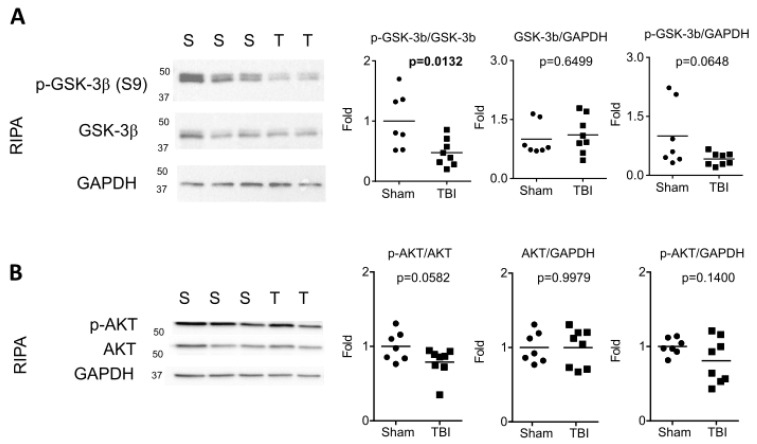
Chronic activation of GSK-3β in rTg4510 after interfaced TBI. Western blotting and quantification of RIPA brain homogenates using antibodies against: (**A**) GSK-3β and p-GSK-3β (S9); and (**B**) AKT and p-AKT (S473). Horizontal lines in graphs indicate group means. The *t*-test was used for all analyses.

**Figure 3 ijms-24-09439-f003:**
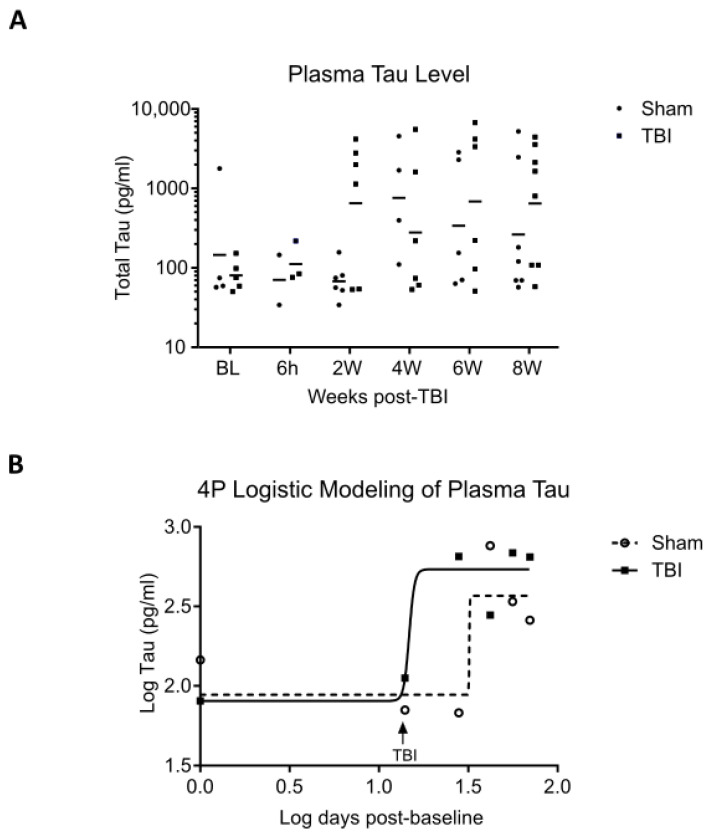
Longitudinal measurement of plasma total tau in rTg4510 mice. (**A**) Baseline (BL) plasma was collected from rTg4510 mice 2 weeks before TBI. Post-TBI plasma was collected at 6 h post-procedure and every 2 weeks thereafter. Plasma total tau was quantified using SIMOA. Horizontal line indicates geometric mean. (**B**) Log plasma total tau vs. log time was modeled using four-parameter logistic (4PL) curves. Symbols overlying the curves represent means of log values. The sham group was best fitted with the equation y = 1.944 + 0.6232/(1 + 10^283.0(1.506−x)^), R^2^ = 0.1959. The TBI group was best fitted with the equation y = 1.905 + 0.8281/(1 + 10^28.91(1.170−x)^), R^2^ = 0.1836.

**Figure 4 ijms-24-09439-f004:**
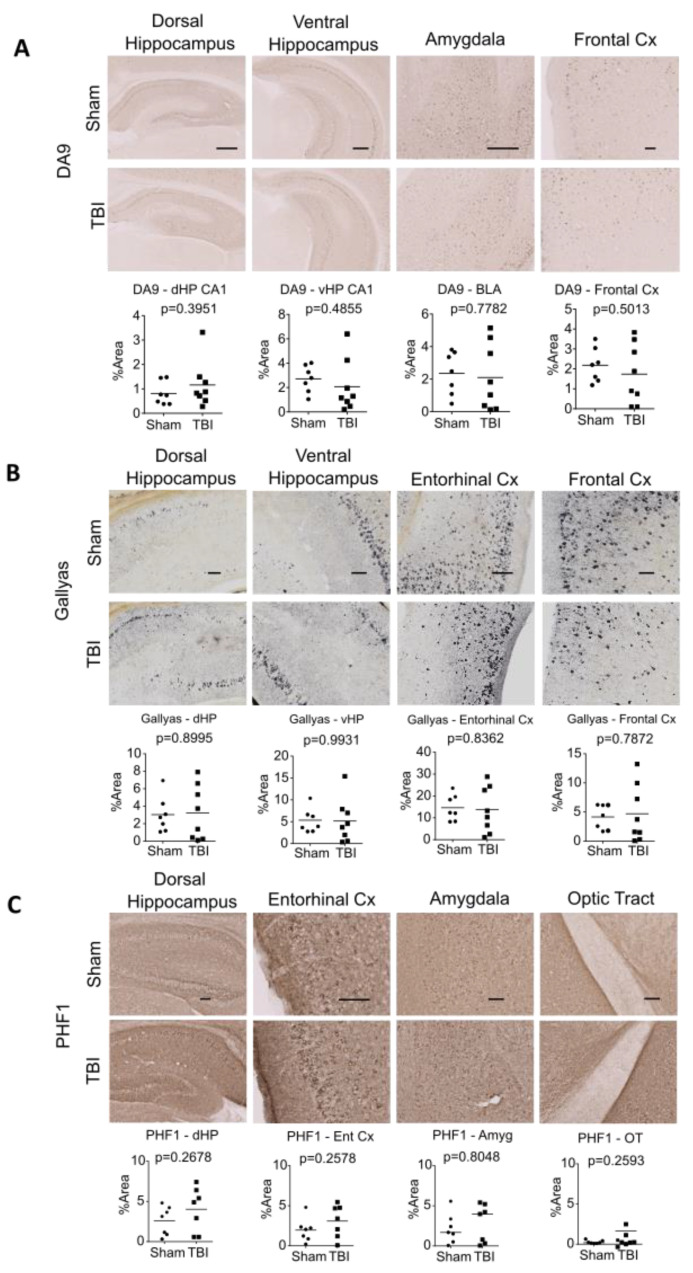
Interfaced TBI at 4.0 J did not change total tau, neurofibrillary tangles, or p-tau immunohistochemistry in rTg4510 mice. (**A**) Immunohistochemistry and quantification of total tau performed using DA9. (**B**) Gallyas Silver staining was performed to visualize neurofibrillary tangles. (**C**–**F**) Immunohistochemistry of p-tau performed using antibodies that recognize different p-tau epitopes: PHF1 (**C**), AT8 (**D**), CP13 (**E**), and MC1 (**F**). Results are quantified in the graphs below the images. Scale bar = 100 μm. The Mann–Whitney U test was used for PHF1-OT, CP13-HP, CP13-Amyg, MC1-Amyg, and MC1-EC, where horizontal lines in graphs indicate group medians. The *t*-test was used for all other analyses, where horizontal lines in graphs indicate group means.

**Figure 5 ijms-24-09439-f005:**
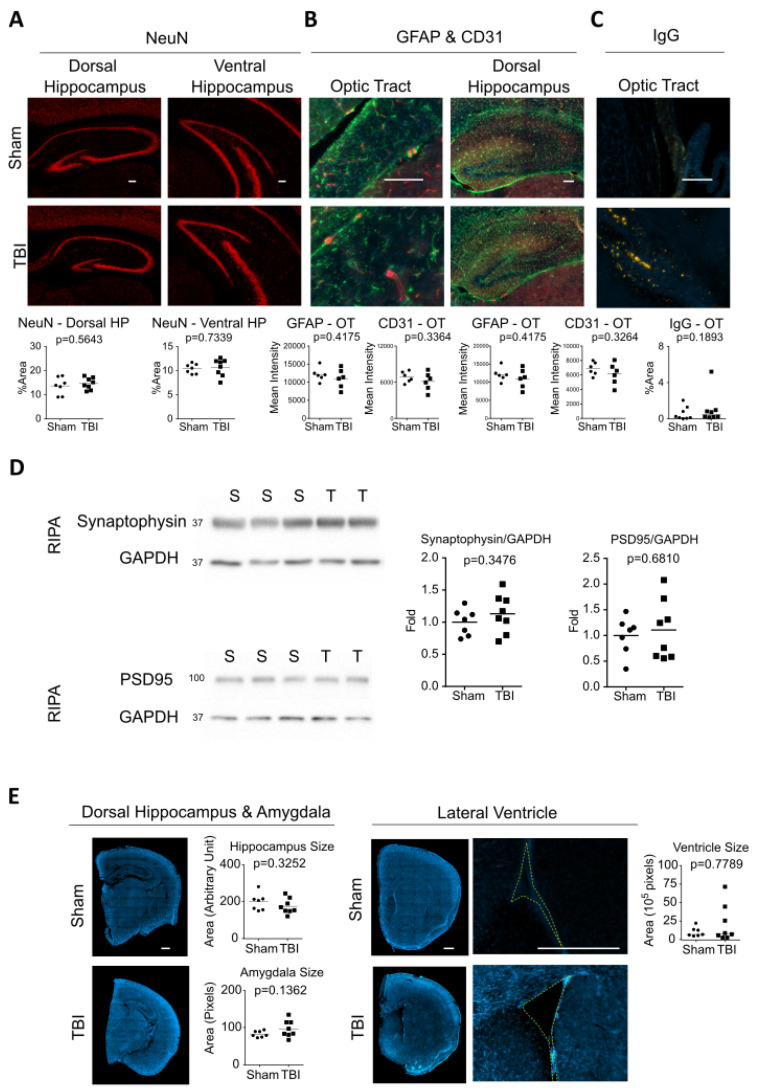
Interfaced TBI at 4.0 J did not change levels of neurons, astrocytes, endothelial cells, or IgG or the synapses or size of brain regions in rTg4510 mice. Immunohistochemistry of (**A**) NeuN (red), (**B**) GFAP (green) and CD31 (red), and (**C**) IgG (yellow) to stain for neurons, astrocytes, endothelial cells and immunoglobulin G, respectively. DAPI (blue) was used as a stain in (**B**,**C**) to visualize cell nuclei. (**D**) Western blotting and quantification of RIPA brain homogenates using antibodies against synaptophysin and PSD95. (**E**) Size comparison of hippocampus, amygdala, and lateral ventricle in sham and TBI mice. Results are quantified in the graphs below the images. Scale bar = 100 μm in (**A**–**C**) and 500 μm in (**E**). The Mann–Whitney U test was used for IgG-OT and ventricle size, where horizontal lines in graphs indicate group medians. Th *t*-test was used for all other analyses, where horizontal lines indicated group mean.

## Data Availability

Image quantification macros are available at https://github.com/tomchengwh/ImageJ-Quantification/tree/main/IJM. Other data presented in this study are available on request from the corresponding author.

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
