# Peer review of "Altered Tau Kinase Activity in rTg4510 Mice after a Single Interfaced CHIMERA Traumatic Brain Injury"

_ijms, 2023, doi:10.3390/ijms24119439_

Round 1

Reviewer 1 Report

The paper investigated the probable changes of tauopathy in a TBI model with human Tau transgenic mice. Given the significant tauopathy and neurodegeneration in rtg4510 time, the authors showed that TBI did not exacerbate these neuronal deficits in these mice. It may be appreciated that there is additional discussion about the increased activity of tau kinase without effect on tauopathy.

Author Response

Thank you for the comments. An additional paragraph has been added in Discussion to discuss the possible factors that contributed to no significant change in tau burden despite increased kinase activity:

“At 2-mo post-injury, we observed no significant change in tau burden. Although we confirmed that tau expression remained unchanged (Supp Fig 2 E-G), we noted that the TBI group showed increased variability across several measures. Specifically, the coefficient of variation (CoV) of tau and p-tau across multiple brain areas in sham mice is ~50% but 90% in the TBI group. Similar observations are found in ventricle sizes. Therefore, it is possible that variations in post-injury tau clearance efficiency may have masked the effects of tau kinase activity. Using Iba1 as microglia marker, we observed no significant differences in the cortical microglial response. Future studies could use phagocytic markers, such as CD68 and Trem2 44,45, to determine if microglia show potential phagocytic differences. In addition, we used p62, LC3B, cathepsin B, and cathepsin D to investigate autophagolysosomes, a major cellular degradation pathway of tau 46-48, and observed no significant differences between sham and TBI animals (Supp Fig 3). Notably, p62 and CTSD level were significantly and positively correlated with cortical tau levels, with a subset of TBI samples with very high p62 levels. (Supp Fig 4). This observation is aligned with previous reports that suggest TBI may impair autophagic flux 49. Future studies could investigate if variations in post-injury autophagy response contribute to tau clearance.”

Reviewer 2 Report

This manuscript describes altered Tau Kinase activity in a mouse model of Traumatic Brain Injury. This manuscript is well written and the results align with the overall goal of the subject. Below are my comments for this manuscript.

1) Fig 5A. Were the same mice used for all the time points until 8 weeks? The total tau levels seem elevated after 2W but then seem to go down for the TBI mice for weeks 4 and 6. If the levels do not drop after 2W then it appears that the total tau levels plateau after 2W without any increase. Is this something that is routinely observed or was it due to loss of material as mentioned in the results section.

2) Since there is no changes in the brain burden at 2mo post injury, do the authors believe that microglia are playing a part in reducing this burden? Do the authors observe changes in microglia activation status?

Author Response

(1) Thank you for the comments. To address the reviewer’s questions, the plasma tau data are replotted below to indicate the longitudinal profile of each mouse (where samples were available), in addition to the 2 plots we included in the manuscript (please see the attached pdf).

We observed that any single mouse may show strong fluctuations in plasma tau levels across 2 consecutive weeks, for reasons unknown. However, if we analyze the group average (mean of log concentration) at each time point, there is a trend toward elevated plasma tau that is accelerated by TBI. This latter analysis is the results that we have reported in the manuscript.

Please see the attachment for the additional figure.

(2) Reviewer 1 has a similar comment as well. We observed no significant differences in the cortical microglial response using Iba1 staining. However, future experiments will be needed to determine if microglia become more variable in their phagocytic activity after TBI. The following paragraph is added to our Discussion section:

“At 2-mo post-injury, we observed no significant change in tau burden. Although we confirmed that tau expression remained unchanged (Supp Fig 2 E-G), we noted that the TBI group showed increased variability across several measures. Specifically, the coefficient of variation (CoV) of tau and p-tau across multiple brain areas in sham mice is ~50% but 90% in the TBI group. Similar observations are found in ventricle sizes. Therefore, it is possible that variations in post-injury tau clearance efficiency may have masked the effects of tau kinase activity. Using Iba1 as microglia marker, we observed no significant differences in the cortical microglial response. Future studies could use phagocytic markers, such as CD68 and Trem2 44,45, to determine if microglia show potential phagocytic differences. In addition, we used p62, LC3B, cathepsin B, and cathepsin D to investigate autophagolysosomes, a major cellular degradation pathway of tau 46-48, and observed no significant differences between sham and TBI animals (Supp Fig 3). Notably, p62 and CTSD level were significantly and positively correlated with cortical tau levels, with a subset of TBI samples with very high p62 levels. (Supp Fig 4). This observation is aligned with previous reports that suggest TBI may impair autophagic flux 49. Future studies could investigate if variations in post-injury autophagy response contribute to tau clearance.”
